# WorldGym: World Model as An Environment for Policy Evaluation

**Julian Quevedo**[1]   **Ansh Kumar Sharma**[2]   **Yixiang Sun**[2]   **Varad Suryavanshi**[2]
**Percy Liang**[1]   **Sherry Yang**[1,2,3]
[1]Stanford University   [2]NYU   [3]Google DeepMind
julianq@stanford.edu, sherryyang@nyu.edu

## Abstract

Evaluating robot control policies is difficult: real-world testing is costly, and handcrafted simulators require manual effort to improve in realism and generality. We propose a world-model-based policy evaluation environment (WorldGym), an autoregressive, action-conditioned video generation model which serves as a proxy to real world environments. Policies are evaluated via Monte Carlo rollouts in the world model, with a vision-language model providing rewards. We evaluate a set of VLA-based real-robot policies in the world model using only initial frames from real robots, and show that policy success rates within the world model highly correlate with real-world success rates. Moreover, we show that WorldGym is able to preserve relative policy rankings across different policy versions, sizes, and training checkpoints. Due to requiring only a single start frame as input, the world model further enables efficient evaluation of robot policies' generalization ability on novel tasks and environments. We find that modern VLA-based robot policies still struggle to distinguish object shapes and can become distracted by adversarial facades of objects. While generating highly realistic object interaction remains challenging, WorldGym faithfully emulates robot motions and offers a practical starting point for safe and reproducible policy evaluation before deployment.[1]

## 1 Introduction

Robots can help humans in ways that range from home robots performing chores (Shafiullah et al., 2023; Liu et al., 2024) to hospital robots taking care of patients (Soljacic et al., 2024). One of the major road blocks in the development robots lies in evaluation — how should we ensure that these robots will work reliably without causing any physical damage when deployed in the real world? Traditionally, people have used *handcrafted* software simulators to develop and evaluate robot control policies (Tedrake et al., 2019; Todorov et al., 2012; Erez et al., 2015). However, handcrafted simulation based on our understanding of the physical world can be limited, especially when it comes to hardcoding complex dynamics with high degrees of freedom or complex interactions such as manipulating soft objects (Sünderhauf et al., 2018; Afzal et al., 2020; Choi et al., 2021). As a result, the sim-to-real gap has hindered progress in robotics (Zhao et al., 2020; Salvato et al., 2021; Dulac-Arnold et al., 2019).

With the development of generative models trained on large-scale video data (Ho et al., 2022; Villegas et al., 2022; Singer et al., 2022), recent work has shown that video world models can visually emulate interactions with the physical real world, by conditioning on control inputs in the form of text (Yang et al., 2023; Brooks et al., 2024) or keyboard strokes (Bruce et al., 2024). This brings up an interesting question — could video world models be used to emulate robot interactions with the real world, hence being used as an environment to evaluate robot policies in the world model before real-world testing or deployment?

Learning a dynamics model from past experience and performing rollouts in the learned dynamics model has been extensively studied in model-based reinforcement learning (RL) (Hafner et al., 2019; Fonteneau et al., 2013; Zhang et al., 2021; Kaiser et al., 2019; Yu et al., 2020). However, most of the existing work in model-based RL considers single-task settings, which puts itself at a disadvantage compared to model-free RL, since learning a dynamics model can be much harder than learning a policy in the single-task setting. Nevertheless, we make the important observation that

---

[1]See videos and code at https://world-model-eval.github.io

> *While there can be many tasks and policies, there is only **one physical world** in which we live that is governed by the **same set of physical laws**.*

This makes it possible to learn a single world model that, in principle, can be used as an interactive environment to evaluate any policies on any tasks.

Inspired by this observation, we propose a world-model-based policy evaluation environment (WorldGym), as shown in Figure 1. WorldGym first combines knowledge of the world across diverse environments by learning a *single* world model that generates videos conditioned on actions. To enable efficient rollouts of policies which predict different-length action chunks, WorldGym aligns its diffusion horizon length with policies' chunk sizes at inference time. With video rollouts from the world model, WorldGym then uses a vision-language model (VLM) to determine tasks' success from generated videos.

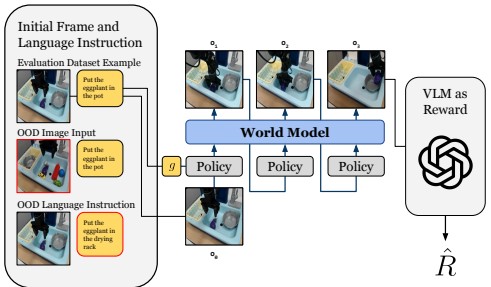

Figure 1: **Overview of WorldGym.** Given an initial frame and an action sequence predicted by a policy, WorldGym uses a world model to interactively predict future frames, serving as a generative simulator. WorldGym then passes the generated rollout to a VLM which provides rewards. WorldGym can easily be used to test policies on OOD tasks and environments by changing the input language instruction or directly modifying the initial image.

Our experiments show that WorldGym can emulate end-effector controls across different control axes highly effectively for robots with different morphologies. We then use the world model to evaluate VLA-based robot policies by rolling out the policies in the world model starting from real initial frames, and compare their success rates (policy values) in WorldGym to those achieved in real-world experiments. Our result suggests that policy values in WorldGym are highly correlated with policy performance in the real world, and the relative rankings of different policies are preserved.

Furthermore, as WorldGym requires only a single initial frame as input, we show how we can easily design out-of-distribution (OOD) tasks and environments and then use WorldGym to evaluate robot policies within these newly "created" environments. We find that modern robot policies still struggle to distinguish some classes of objects by their shape, and can even be distracted by adversarial facades of objects.

Although simulating realistic object interactions remains challenging, we believe WorldGym can serve as a highly useful tool for sanity check and testing robot policies safely and reproducibly before deploying them on real robots. Key contributions of this paper include:

- We propose to use video world model to evaluate robot policies across different robot morphologies, and perform a comprehensive set of studies to understand its feasibility.
- We propose flexibly aligning diffusion horizon length with policies' action chunk sizes for efficient rollouts of a variety of policies over hundreds of interactive steps.
- We show a single world model learned on data from diverse tasks and environments can enable policy value estimates that highly correlate with real-world policy success rates.
- We demonstrate the ease of testing robot policies on OOD tasks and environments within an autoregressive video generation-based world model.

## 2 PROBLEM FORMULATION

In this section, we define relevant notations and review the formulation of offline policy evaluation (OPE). We also situate OPE in practical settings with partially observable environments and image-based observations.

**Multi-Task POMDP.** We consider a multi-task, finite-horizon, partially observable Markov Decision Process (POMDP) (Puterman, 2014; Kaelbling et al., 1995), specified by $\mathcal{M} = (S, A, O, G, R, T, \mathcal{E}, H)$, which consists of a state space, action space, observation space, goal space, reward function, transition function, emission function, and horizon length. A policy $\pi$ interacts with the environment for a goal starting from an initial state $g, s_0 \sim G$, producing a distribution $\pi(\cdot|s_t, g)$ over $A$ from which an action $a_t$ is sampled and applied to the environment at each step $t \in [0, H]$. The environment produces a scalar reward $r_t = R(s_t, g)$, and transitions to a new state $s_{t+1} \sim T(s_t, a_t)$ and emits a new observation $o_{t+1} \sim \mathcal{E}(s_{t+1})$. We consider the sparse reward setting

with $R(s_H, g) \in \{0, 1\}$ and $R(s_t, g) = 0, \forall t < H$, where $g$ is a language goal that defines the task. Data is logged from previous interactions into an offline dataset $D = \{g, s_0, o_0, a_0, ..., s_H, o_H, r_H\}$. The value of a policy $\pi$ can be defined as the total expected future reward:

$$\rho(\pi) = \mathbb{E}[R(s_H, g)|s_0, g \sim G, a_t \sim \pi(s_t, g),$$
$$s_{t+1} \sim T(s_t, a_t), \forall t \in [0, H]]. \tag{1}$$

Estimating the value of $\rho(\pi)$ from previously collected data $D$, known as offline policy evaluation (OPE) (Levine et al., 2020), has been extensively studied (Thomas & Brunskill, 2016; Jiang & Li, 2016; Fu et al., 2021; Yang et al., 2020; Thomas et al., 2015b). However, existing work in OPE mostly focuses on simulated settings that are less practical (e.g., assumptions about full observability, access to ground truth states).

**Model-Based Evaluation.** Motivated by characteristics of a real-robot system such as image based observations, high control frequencies, diverse offline data from different tasks/environments, and the lack of access to the ground truth state of the world, we consider the use of offline data to learn a *single* world model $\hat{T}(\cdot|\mathbf{o}, \mathbf{a})$, where $\mathbf{o}$ represents a sequence of previous image observations and $\mathbf{a}$ represents a sequence of next actions. A sequence of next observations can be sampled from the world model $\mathbf{o}' \sim \hat{T}(\mathbf{o}, \mathbf{a})$. With this world model, one can estimate the policy value $\rho(\pi)$ with Monte-Carlo sampling using stochastic rollouts from the policy and the world model:

$$\hat{\rho}(\pi) = \mathbb{E}[\hat{R}([o_0, ..., o_H], g)|s_0, g \sim G, \mathbf{a} \sim \pi(\mathbf{o}, g),$$
$$\mathbf{o}' \sim \hat{T}(\mathbf{o}, \mathbf{a}), \mathbf{o} = \mathbf{o}'], \tag{2}$$

where $\hat{R}$ is a learned reward function. Previously, model-free policy evaluation may be more preferable since in a single task setting, dynamics models are potentially harder to learn than policy values themselves, and doing rollouts in a dynamics model may lead to compounding errors (Xiao et al., 2019). However, we make the key observations that while there can be many tasks and many policies, there is only one physical world that is governed by the same set of physical laws. As a result, learning a world model can benefit from diverse data from different tasks and environments with different state spaces, goals, and reward functions. More importantly, a world model can be directly trained on image-based observations, which is often the perception modality of real-world robots.

## 3 BUILDING AND EVALUATING THE WORLD MODEL

In this section, we first describe our implementation of world model training and inference. Then, we discuss how we validate our world model's performance prior to rolling out real robot policies within it in the next section.

### 3.1 BUILDING THE WORLD MODEL

First, we describe the architecture and key implementation details, followed by our proposed inference scheme for policy rollouts.

#### 3.1.1 WORLD MODEL TRAINING

We train a latent Diffusion Transformer (Peebles & Xie, 2023) on sequences of frames paired with actions, using Diffusion Forcing (Chen et al., 2024) to enable autoregressive frame generation. Per-frame robot action vectors are linearly projected to the model dimension and added elementwise to diffusion timestep embeddings, the result of which is used to condition the model through AdaLN-Zero modulation, similar to class conditioning in Peebles & Xie (2023). To ensure the world model is fully controllable by robot actions, we propose to randomly drop out actions for entire video clips, and use classifier-free guidance to improve the world model's adherence to action inputs. Conditioning on previous frames' latents is achieved via causal temporal attention blocks interleaved between spatial attention blocks, as in Bruce et al. (2024); Ma et al. (2025). See Appendix A for additional implementation details.

#### 3.1.2 ROLLING OUT A POLICY IN THE WORLD MODEL

Our policy evaluation pipeline operates through an iterative loop between the robot policy and the world model. First, the world model is initialized with an initial observation $o_0$, which is then passed as input to a policy $\pi$ which produces a chunk of actions $\mathbf{a}_{\text{pred}}$. The actions are passed back to the world model, which predicts a new frame for each action in $\mathbf{a}_{\text{pred}}$. The latest frame produced by the world model is then returned to the policy as its next input observation.

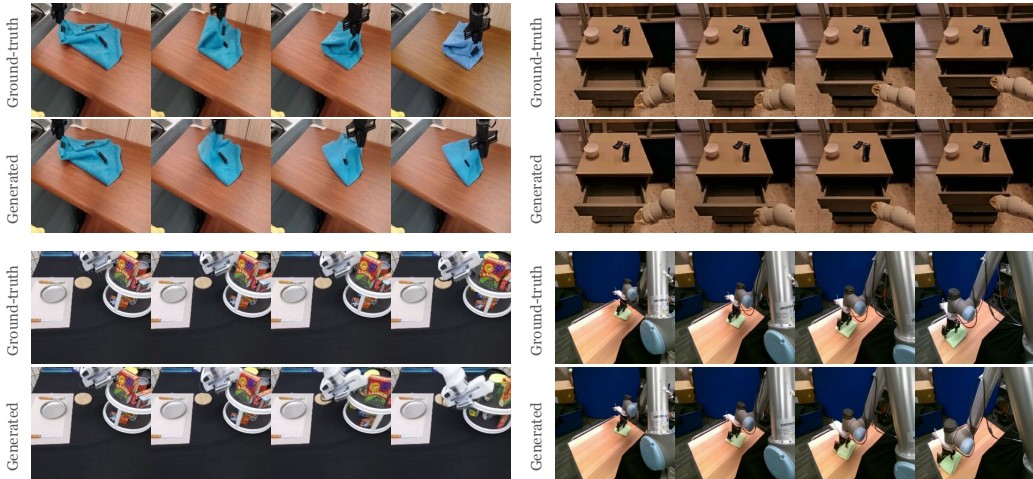

Figure 2: **Qualitative evaluation of the world model on Bridge, RT-1, VIOLA, and Berkeley UR5.** In each group, top row shows the ground truth video from the real robot. Bottom row shows the generated video from the world model conditioned on the *same* actions as the original video. The world model closely follows the true dynamics across different robot morphologies.

Since different robot policies output a different number of actions at once (Kim et al.; Brohan et al., 2022; Chi et al., 2023), WorldGym needs to support efficient prediction of a chunk of videos conditioned on a chunk of (variable length) actions. By virtue of being trained with Diffusion Forcing, as well as our usage of a causal temporal attention mask, we can flexibly control how many frames our world model denoises in parallel at inference time, i.e. its prediction *horizon length*. We propose setting the horizon equal to the policy's action chunk size, $|\mathbf{a}_{\text{pred}}|$. This has the benefit of efficient frame generation for policies with differing action chunk sizes, all from a single world model checkpoint. This contrasts with prior diffusion world models for robotics, such as Cosmos (NVIDIA et al., 2025), which, due to being trained with bidirectional attention and a fixed context length, must always denoise 16 latent frames in parallel. This constraint results in wasted compute for action chunk sizes less than the context length and unrealized parallelism for chunk sizes which are larger. On the other hand, our design allows parallelism to flexibly match the number of actions, thus utilizing hardware more effectively (see Appendix F.2).

### 3.1.3 VLM AS REWARD

We opt for GPT-4o (OpenAI et al., 2024) as a reward model, passing in the sequence of frames from the generated rollout and the language instruction (see the prompt for the VLM in Appendix B). In certain cases where both policies being evaluated fail to perform a task end-to-end, it is still helpful to get signals on which policy is closer to completing a task. We can specify these partial credit criteria to the VLM to further distinguish performance between different policies, which has been done manually using heuristics in prior work (Kim et al.). We validate the accuracy of VLM-predicted rewards in Appendix B.2.

### 3.2 EVALUATING THE WORLD MODEL

Next, we describe how we validate the performance of our world model prior to policy evaluation, ensuring that it exhibits realistic robot movement and adheres to arbitrary action controls.

### 3.2.1 AGREEMENT WITH VALIDATION SPLIT

First, we test the world model's ability to generate similar videos as running a robot in the real world. Specifically, we take the validation split of initial images from the Open-X Embodiment dataset, and predict videos conditioned on the *same* action sequences as in the original data. Figure 2 shows that the generated rollouts generally follow the real-robot rollouts across different initial observations and different robot morphologies.

### 3.2.2 END-EFFECTOR CONTROL SWEEPS

Next, we need a way to evaluate whether our world model can emulate arbitrary action sequences, beyond the kinds of action sequences present in the training data. We propose hard-coding a robot

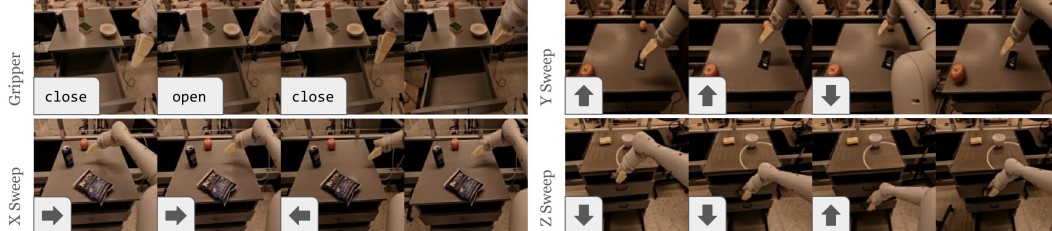
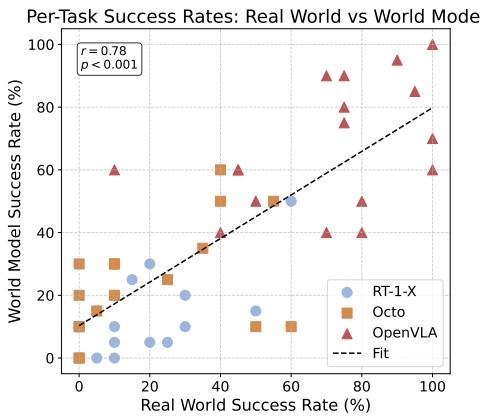
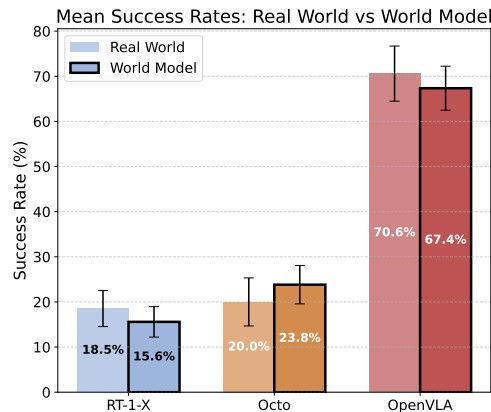

Figure 3: **Results on end-effector control** across action dimensions. Generated videos closely follow the gripper controls such as open and close the gripper as well as moving in different directions starting from any initial observation frame. Results for control sweeps on the Bridge robot can be found in Figure 16 in Appendix E.1.

(a) **Per-Task Task Success Rates.** Each point represents a task from Table 5, with different policies being represented by different shaped markers. There is a strong correlation ($r = 0.78$) between policy performance in our world model (y-axis) and within the real world (x-axis).

(b) **Mean Success Rates.** Robot policies' mean success rates in the world model differ by an average of only 3.3% between from the real world, near the standard error range for each policy. Relative performance rankings between RT-1-X, Octo, and OpenVLA are also preserved.

Figure 4: Success rates of modern VLAs, as evaluated within WorldGym and the real world.

control policy by only moving one action dimension at once (and keeping the other action dimensions as zeros). The robot is then expected to move along that one action dimension with non-zero input, corresponding to moving in different horizontal and vertical directions as well as open and close its gripper. Figure 3 shows that the generated videos faithfully follow the intended end-effector movement,[2] despite the fact that these particular sequences of controls are not present in the training data.

## 4 EVALUATING POLICIES IN WORLDGYM

Having established confidence in the world model's performance, we now use the world model to evaluate policies. We begin by rolling out three recent VLA policies in WorldGym and check whether WorldGym **reflects real-world success**. (Section 4.1). We then assess whether **relative policy performance** is preserved, comparing different versions, sizes, and training stages of the same models (Section 4.2). Finally, we explore WorldGym's potential to test policies on **out-of-distribution (OOD) tasks and environments** (Section 4.3), including novel instructions and altered visual contexts.

### 4.1 CORRELATION BETWEEN REAL-WORLD AND SIMULATED POLICY PERFORMANCE

**Qualitative Evaluation.** To ensure WorldGym is useful for policy evaluation, we test whether policy performance within the world model is similar to that of the real world. To do so, we perform a direct comparison with the Bridge evaluation trials from OpenVLA (Kim et al.). Specifically, the OpenVLA Bridge evaluation consists of 17 challenging tasks which are not present in the Bridge V2 (Walke et al., 2023) dataset. We use WorldGym to evaluate the three open-source policies evaluated

---

[2]Results are best viewed as videos in the supplementary material.

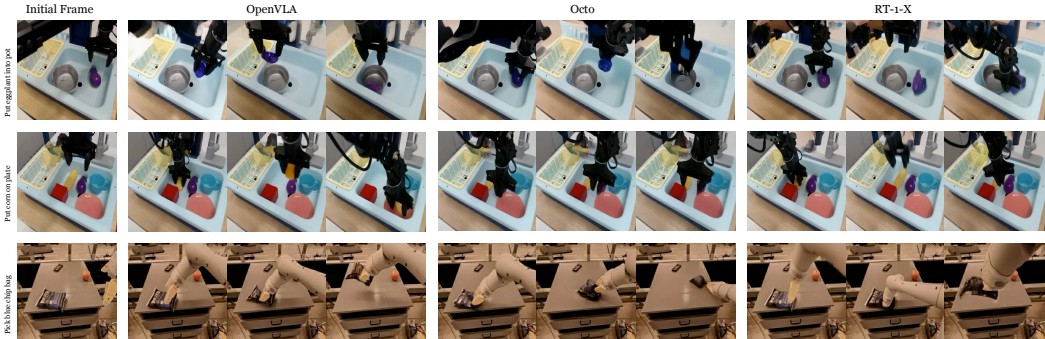

Figure 5: **Qualitative policy rollouts on Bridge and Google Robot** for RT-1-X, Octo, and OpenVLA. OpenVLA rollouts often lead to more visual successes than the other two policies across environments.

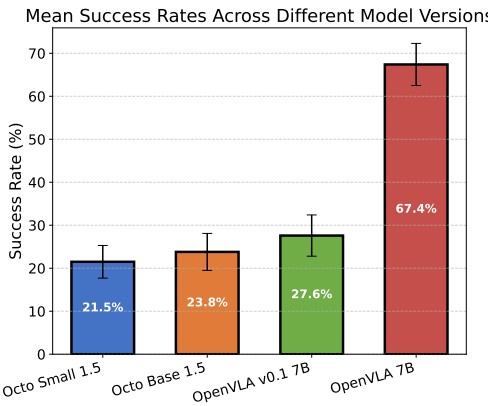

Figure 6: **Success Rates of different model versions in WorldGym.** We evaluate different generations of Octo and OpenVLA in the world model, showing that WorldGym assigns higher score to larger and more recent versions.

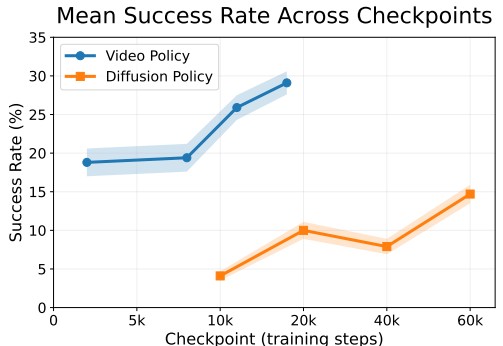

Figure 7: **Success Rate within WorldGym throughout training.** We train a video-based policy and a diffusion policy from scratch and evaluate it within our world model as it trains. We see that mean task success rate within the world model increases with additional training steps.

in Kim et al.: RT-1-X (O'Neill et al., 2023), Octo (Octo Model Team et al., 2024), and OpenVLA (Kim et al.). For each task and each policy, Kim et al. perform 10 trials, each with randomized initial object locations. We obtain the first frame of the recorded rollouts for all trials of all tasks. We then simulate each of the 10 real-world trials by using the original initial frame to roll out the policy within the world model as described in Section 3.1.2. We show qualitative rollouts in WorldGym from different policies in Figure 5, which shows that rollouts from OpenVLA generally perform better than rollouts from RT-1-X and Octo on the Bridge robot (top two rows). We further show that WorldGym can be easily used to perform rollouts in other environments with other robots, such as the Google Robot (bottom row in Figure 5).

**Quantitative Evaluation.** Using the simulated rollouts from WorldGym, we then compute the average task success rate similar to Kim et al., and plot the success rate for each task for each policy in Figure 4a. We find that real-world task performance is strongly correlated with the task performance reported by the world model, achieving a Pearson correlation of $r = 0.78$. While per-task policy success rates within WorldGym still differ slightly from those in the real world (see Table 5), the mean success rates achieved by these policies within WorldGym are quite close to the their real-world values, as shown in Figure 4b. The success rates differ by an average of only 3.3%, with RT-1-X achieving 18.5% in the real world vs 15.5% in the world model, Octo achieving 20.0% vs 23.82%, and OpenVLA achieving 70.6% vs 67.4%, respectively. See quantitative results of evaluating the three policies on the Google Robot in Appendix E.2

## 4.2 Policy Ranking within a World Model

Now we test whether WorldGym can preserve policy rankings known a priori. We evaluated policies across different versions, sizes, and training stages within WorldGym on the OpenVLA Bridge

evaluation task suite, and found their in-world-model performance rankings to be consistent with prior knowledge of their relative performance.

**Different VLAs with Known Ranking.** First, we average success rates across all 17 tasks and find that the relative performance rankings between RT-1-X, Octo, and OpenVLA are the same (Figure 4b) within both WorldGym and the real-world results reported in OpenVLA (Kim et al.).

**Same Policies across Versions and Sizes.** We further examine whether WorldGym preserves rankings between different versions and sizes of the same policy. In particular, we compare Octo-Small 1.5 against Octo-Base 1.5, and OpenVLA v0.1 7B, an undertrained development model, against OpenVLA 7B. As shown in Figure 6, the larger and more recent models outperform their smaller or earlier counterparts within WorldGym, consistent with the findings of real-world experiments performed in Octo Model Team et al. (2024) and Kim et al.. This provides additional evidence that WorldGym faithfully maintains relative rankings even across model upgrades.

**Same Policy across Training Steps.** To examine whether WorldGym provides meaningful signals for policy training, hyperparameter tuning, and checkpoint selections, we train two robot policies from scratch. Building on prior evidence of WorldGym's effectiveness in evaluating VLA-based policies, we extend our study to two additional families: a video prediction–based policy (UniPi) (Du et al., 2023a) and a diffusion-based policy (DexVLA) (Wen et al., 2025), both trained on the Bridge V2 dataset (see Appendix C and Appendix D). We evaluate checkpoints of the video prediction policy at 2K, 8K, 12K, and 18K steps, and the diffusion policy at 10K, 20K, 40K, and 60K steps.

As shown in Figure 7, WorldGym tends to assign higher success rates to checkpoints as they increase in training steps, consistent with the lower mean squared error these policies achieve on their validation splits. This demonstrates WorldGym's ability to preserve policy rankings across models with different amounts of training compute.

Thus, we have shown how WorldGym can be used to obtain reasonable policy rankings. In particular, for the VLA-based policies we evaluate, we arrive at the same conclusions as real-world experiments about their relative performances. Notably, this is achieved all **without the manual effort** of setting up real robot evaluation environments and monitoring policy rollouts. While real-world evaluation can sometimes take days to complete, all WorldGym rollouts reported here can be completed in under an hour on a single GPU and require only initial images for each trial.

## 4.3 OUT-OF-DISTRIBUTION INPUTS

In this section, use WorldGym to explore policies' performance on both OOD input images and OOD language instructions.

**OOD Image Input.** Using modern image generation models like Nano Banana (Google, 2025), we can easily generate new input images to initialize our world model with. We evaluate robot policies under three OOD settings: unseen object interaction, distractor objects, and object classification (see detailed results in Table 6).

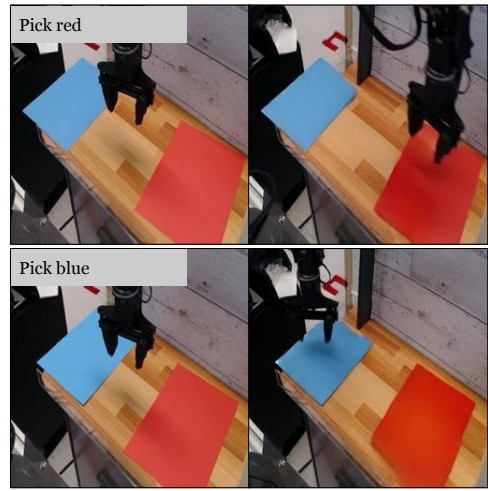

- **Unseen Objects:** We edit a scene to contain both a carrot and an orange, asking the policy to pick up the orange (Figure 9). OpenVLA grabs whichever object is closer until we edit the carrot's color to be red, after which it always grabs the orange correctly. This suggests that it struggles to distinguish carrots and oranges by their shape.
- **Distractor Objects:** We use the image editing model to add a computer displaying an image of a carrot (Figure 10, left). We see that OpenVLA mistakenly to grabs the carrot on the computer screen in 15% of trials, suggesting limited 3D/2D object distinction.
- **Classification:** We add a piece of paper on each side of a desk. We first color one paper red and the other blue and instruct the model to "pick red"/"pick blue" (Figure 8). OpenVLA achieves

Figure 8: **OOD: Color Classification.** We add red and blue pieces of paper to a table, and ask the policies to "pick red" or "pick blue" (OOD image and language). OpenVLA excels, picking the correct colored paper in all trials, whereas all other policies score near chance.

a perfect score, always moving towards the

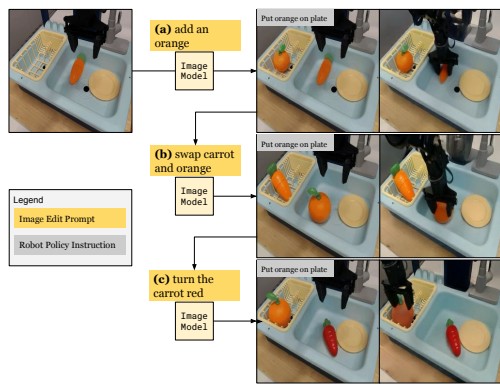 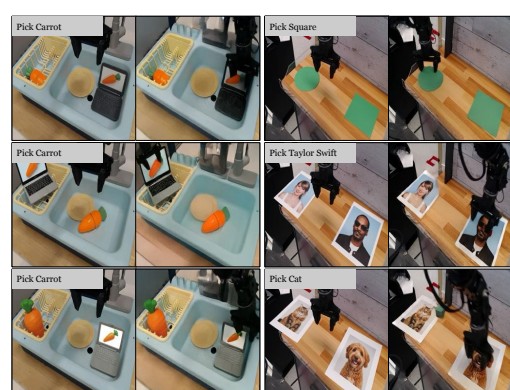

Figure 9: **OOD: Unseen object.** We use Nano Banana (Google, 2025) to add an orange to the world model's initial frame. When both the orange and the carrot are present, (a-b) OpenVLA grabs whichever is closer. After (c) editing the carrot's color to red, however, the orange is correctly picked up.

Figure 10: **OOD: Failure modes.** *Left:* We add a laptop to the scene, which displays an image of a carrot. In 15% of trials, OpenVLA grabs the laptop instead of the real carrot. *Right:* We test the ability distinguish to between squares and circles, celebrity faces, and cats and dogs, with all policies scoring near-chance.

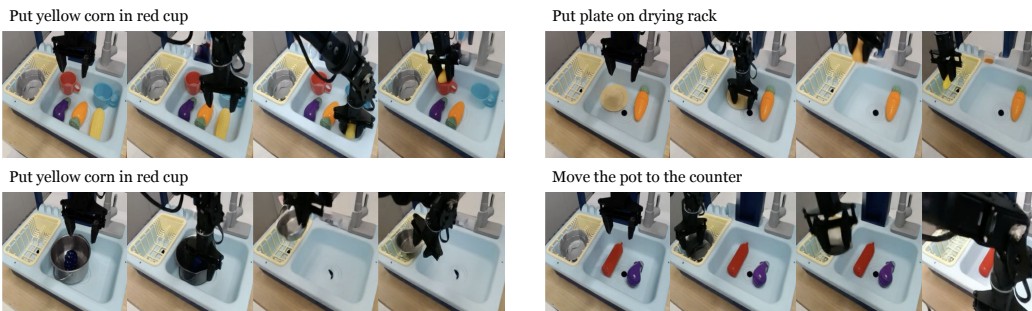

Figure 11: **OOD Language Instructions.** We pick a set of tasks from the OpenVLA Bridge evaluation suite and modify the language instruction, e.g. changing the the target object and/or its goal destination.

correct color. Octo and RT-1, on the other hand, typically move towards whichever paper is closer, scoring no better than chance. We also try more advanced classification tasks, (Figure 10, right), but find that the policies all score near-chance.

For a more quantitative study, we modify all the initial frames of the OpenVLA's Bridge evaluation task suite to include random OOD distractor items (see Figure 12), keeping the language instructions the same. We then repeat the rollout procedure from Section 4.1 in order to measure the degree to which the addition of unrelated objects affects policy performance. We find that all the tested VLAs degrade in performance, with OpenVLA being the most robust of the three (Figure 13).

**OOD Language.** Additionally, even without access to an image editing model, we demonstrate that WorldGym can be used to evaluate policies' performance on OOD language instructions. Starting from a set of initial frames from the tasks listed in Table 5, we modify each task's language instruction, e.g. changing the target object and/or its goal state. Figure 11 shows rollouts from OpenVLA for these OOD language tasks. We can then easily obtain success rates for these unseen tasks by rolling them out within WorldGym, finding that OpenVLA generalizes best (see Table 1). Policies struggle across the board on the "Move the pot to the counter" task, with only OpenVLA achieving a single success. We suspect that OpenVLA consistently outperforms Octo and RT-1-X on OOD language tasks due to its strong VLM backbone and richer robot pretraining dataset (Kim et al.).

The ability to use WorldGym to quickly design and evaluate policies within OOD tasks and environments thus leads us to new findings about policies' strengths and weaknesses. Future research could be prioritized to address these issues, all without spending extra effort to set up additional experiments in the real world or within handcrafted simulators.

| Task | RT-1-X | Octo | OpenVLA |
|---|---|---|---|
| Move Pot Into Drying Rack | 3 | 0 | 7 |
| Move The Pot To The Counter | 0 | 0 | 1 |
| Put Plate On Drying Rack | 4 | 2 | 8 |
| Put Yellow Corn In Red Cup | 1 | 2 | 3 |

Table 1: **Policy Evaluations Results on Bridge OOD Language Tasks.** "Move the pot to the counter" is perhaps the most challenging because the Bridge dataset does not contain trajectories which move objects outside of the sink basin. OpenVLA has the strongest performance, which we attribute to its more powerful language model backbone.

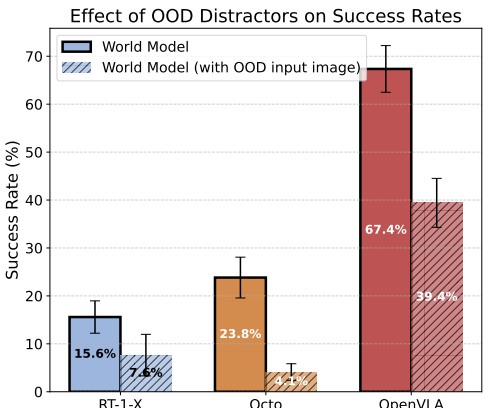

Figure 13: **Effect of OOD Distractors.** We use an image editing model to add distractor objects to the Bridge evaluation suite, finding that RT-1-X drops in performance by 51%, Octo by 83%, and OpenVLA by 41.5%, making OpenVLA the most robust to distractors. See Table 7 for details.

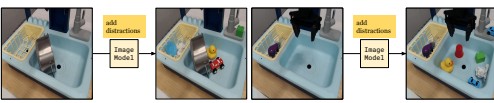

Figure 12: **OOD Distraction Examples.** We use Nano Banana (Google, 2025) to add distractions to every image of the OpenVLA Bridge task suite. The resulting change in mean success rates can be seen in Figure 13.

## 5 RELATED WORK

**Action-Conditioned Video Generation.** Previous work has shown that video generation can simulate real-world interactions (Yang et al., 2023; Brooks et al., 2024), robotic plans (Du et al., 2024; 2023b), and games (AI et al., 2024; Bruce et al., 2024; Valevski et al., 2024; Alonso et al., 2024) when conditioned on text or keyboard controls. Prior work (NVIDIA et al., 2025) has begun to explore applying video generation to simulating complex robotic controls. We take this a step further by using video-based world models to quantitatively estimate robot policy success rates. WorldGym draws architectural inspirations from prior work on video generation such as Diffusion Forcing (Chen et al., 2024) and Diffusion Transformers (Peebles & Xie, 2023), but experiments with variable horizon lengths to support efficient long-horizon rollouts for policies with a variety of action chunk sizes.

**Policy Evaluation.** Off-policy and offline policy evaluation has long been studied in the RL literature (Farajtabar et al., 2018; Jiang & Li, 2015; Kallus & Uehara, 2019; Munos et al., 2016; Precup et al., 2000; Thomas et al., 2015a). Some of these approaches are model-based, learning a dynamics model from previously collected data and rolling out the learned dynamics model for policy evaluation (Fonteneau et al., 2013; Zhang et al., 2021; Yu et al., 2020; Hafner et al., 2020). Since learning a dynamics model is challenging and subject to accumulation of error, a broader set of work has focused on model-free policy evaluation, which works by estimating the value function (Le et al., 2019; Duan & Wang, 2020; Sutton et al., 2009; 2016) or policy correction (Kanamori et al., 2009; Nguyen et al., 2010; Nachum et al., 2019). WorldGym performs model-based policy evaluation, but proposes to learn a single world model on image-based observation that can be used to evaluate different policies on different tasks. SIMPLER (Li et al., 2024) aims to evaluate realistic policies by constructing software-based simulators from natural images and showed highly correlated curves between simulated evaluation and real-robot execution, but it is hard to evaluate OOD language and image input in SIMPLER without significant hand engineering of the software simulator. Li et al. (2025) proposes to evaluate robot policies in a world model in a specific bi-manual manipulation setup, whereas WorldGym focuses on evaluating policies across diverse environments and robot morphologies while enabling testing OOD language and image inputs.

## 6 CONCLUSION

We have presented WorldGym, a world-model-based environment for evaluating robot policies. WorldGym emulates realistic robot interactions and shows strong correlations between simulated evaluation and real-world policy outcomes. WorldGym further provides the flexibility for evaluating OOD language instructions and performing tasks with an OOD initial frame. While not all interactions emulated by WorldGym are fully realistic, WorldGym serves as an important step towards safe and reproducible policy evaluation before deployment.

ACKNOWLEDGMENTS

We thank Xinchen Yan and Doina Precup for reviewing versions of this manuscript. We thank Moo Jin Kim for help in setting up the OpenVLA policy. We thank Boyuan Chen and Kiwhan Song for the Diffusion Forcing GitHub repository.

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

# Appendix

## A ADDITIONAL DETAILS OF THE AUTOREGRESSIVE DIFFUSION TRANSFORMER

**Implementation details:** We use the VAE from Stable Diffusion 3 Esser et al. (2024) to independently encode 256×256 image frames into latent space. We employ a 16-layer transformer with 1024 hidden dimensions and 16 attention heads. We train the world model on a diverse set of data sources, including 9 of the robot datasets from Open-X Embodiment whose action spaces can be unified, such as Bridge V2 (Walke et al., 2023) and RT-1 (Brohan et al., 2022). We encode actions from a 7-dimensional vector, using the 6-dimensional end-effector position and binary gripper state as our action space. Action spaces from different robots are aligned by normalizing each component's 10th- and 90th-percentile values to those of the RT-1 dataset. We train with a context length of 20 frames; for longer rollouts, we condition on a sliding window of the last 20 frames.

| Hyperparameter | Value |
|---|---|
| Total parameters | 609 M |
| Image Resolution | 256×256 |
| DiT Patch Size | 2 |
| Input Channels | 16 |
| Hidden Size | 1024 |
| Layers | 16 |
| Attention Heads | 16 |
| MLP Ratio | 4 |
| Optimizer | AdamW (weight decay = 0.002, $\beta_1 = 0.9$, $\beta_2 = 0.99$) |
| Learning rate | 8e-5 |
| Batch size | 16 |
| Action dimension | 7 |
| Training hardware | 2xA100 80GB |
| Training steps | 300k |
| Diffusion noise schedule | sigmoid |
| Sampling timesteps | 10 |
| Prediction target | $v$ |

Table 2: Hyperparameters for training WorldGym's video prediction model.

---

**Algorithm 1** WorldGym policy evaluation loop.

---

**Require:** World model $\hat{T}$ with training context length $N_{\text{train}}$ and prediction horizon $h$, rollout length $N_{\text{rollout}}$, policy $\pi$ with action chunk size $|\mathbf{a}_{\text{pred}}|$, reward model $\hat{R}$, initial observation $o_0$, goal $g$

$\quad \mathbf{o} \leftarrow [o_0]$

$\quad \mathbf{a} \leftarrow [a_{\text{null}}]$

$\quad n = 0$

$\quad$ **while** $n \leq N_{\text{rollout}}$ **do**

$\quad\quad \mathbf{a}_{\text{pred}} \leftarrow \pi(\mathbf{o}_n, g)$

$\quad\quad$ **for** $i = 0$ to $\lceil |\mathbf{a}_{\text{pred}}|/h \rceil - 1$ **do**

$\quad\quad\quad \mathbf{a}_{\text{ctx}} \leftarrow \mathbf{a}_{-N_{\text{train}}:}$

$\quad\quad\quad \mathbf{o}_{\text{ctx}} \leftarrow \mathbf{o}_{-N_{\text{train}}:}$

$\quad\quad\quad \mathbf{o}_{\text{pred}} \leftarrow \hat{T}(\mathbf{o}_{\text{ctx}}, \mathbf{a}_{\text{ctx}} || \mathbf{a}_{\text{pred}, h \cdot i : h \cdot (i+1)}) \quad\quad\quad\quad \triangleright$ predict a block of $h$ frames in parallel

$\quad\quad\quad \mathbf{o} \leftarrow \mathbf{o} || \mathbf{o}_{\text{pred}} \triangleright$ concatenate generated block of observation frames with observation history

$\quad\quad\quad \mathbf{a} \leftarrow \mathbf{a} || \mathbf{a}_{\text{pred}, h \cdot i : h \cdot (i+1)}$

$\quad\quad$ **end for**

$\quad\quad n \leftarrow n + n_{\text{chunk}}$

$\quad$ **end while**

$\quad r \leftarrow \hat{R}(\mathbf{o})$

---

Algorithm 1 shows the detailed algorithm for performing a sliding window rollout of a policy with action chunk size $|\mathbf{a}_{\text{pred}}|$ and a world model with prediction horizon $h$. Note that in practice we always choose $h = |\mathbf{a}_{\text{pred}}|$ at inference time.

## B    DETAILS OF VLM AS REWARD

### B.1    PROMPT FOR VLM AS REWARD

Prompt GPT-4o as Reward $\hat{R}$. Note that `has_partial` is `True` if the chosen task has a partial credit criteria, which is the case for some tasks used in OpenVLA (Kim et al.).

```
Here is a sequence of frames from a robot policy which has
    been rolled out in a video-generation-based world model.
I need your help determining whether the policy is successful
    . How successfully does the robot complete the following
    task?

Instruction: {instruction}
{rubric.strip()}

Provide brief reasoning (2-3 sentences). Then output EXACTLY
    one final line:
Final Score: X
Where X is { 'one of 0, 0.5, or 1' if has_partial else '0 or
    1' }.
No extra numbers after that line.
Note: Since this video was generated by a video prediction
    model (conditioned on robot actions), it may contain some
     artifacts due to the video model capacity.
```

If there is a partial credit criteria, the rubric is:

```
0   = Failure: little or no progress toward: "{instruction}"
0.5 = Partial: "{partial_desc}" achieved BUT the instruction
    not fully completed
1   = Success: Instruction fully completed (counts even if
    partial also true)
```

Otherwise, the rubric is:

```
Score rubric:
0 = Failure: instruction "{instruction}" not completed.
1 = Success: instruction completed.
```

### B.2    VALIDATING VLM SUCCESS PREDICTIONS

To determine whether a VLM can serve as a reliable reward function, we pass rollout videos from the RT-1 dataset, along with the prompts constructed from the templates above, as inputs to query GPT-4o. We use whether the task is successful according to the RT-1 data (validation split) as the ground truth. Table 3 shows that GPT-4o achieves high true positive and true negative rate for real

Table 3: **Performance of VLM as reward** (mean and standard error across 4 runs) on videos from RT-1 (Brohan et al., 2022) using ground truth task success labels. GPT-4o achieves high true positives and true negatives. Notably, GPT-4o as reward has very low false positive rate, which is especially important for not over-estimating a policy value.

|  | RT-1 Success | RT-1 Fail |
|---|---|---|
| VLM Success | $0.81 \pm 0.14$ (TP) | $0.03 \pm 0.05$ (FP) |
| VLM Fail | $0.19 \pm 0.14$ (FN) | $0.97 \pm 0.05$ (TN) |

videos, indicating that it is an effective evaluator of task success. Notably, GPT-4o achieves very low false positives (i.e., the rollout is a failure but the VLM thinks it is a success), which is highly useful in policy evaluation.

# C ARCHITECTURE AND TRAINING DETAILS OF VIDEO BASED POLICY

Our video-based policy follows the framework of UniPi Du et al. (2023a), combining a language-conditioned video prediction model with an inverse dynamics model.

The video prediction module shares the same architecture as our world model, but replaces the conditioning on robot actions at each timestep with language instructions. For language conditioning, we employ the pretrained and frozen UMT5-xxl encoder Chung et al. (2023) to obtain token-level embeddings. These embeddings are aggregated via mean pooling to form a 4096-dimensional instruction representation. This representation is projected to match the model dimensionality and is used to modulate the diffusion transformer through adaptive layer normalization (adaLN-Zero). In this way, task semantics are directly integrated into the video prediction process. We train our video generation model for 180k steps on Bridge V2 (Walke et al., 2023). The visualization of the video generation policy on validation scenes can be seen in Figure 14.

The inverse dynamics model predicts the action sequence given a short video clip of 10 frames. Each frame is encoded with a ResNet 50 backbone He et al. (2015), producing per-frame features $f_t$. To capture motion, we compute both $f_t$ and temporal differences $\Delta f_t = f_{t+1} - f_t$, concatenate them, and flatten across the clip. The resulting representation is passed through an MLP to predict $10 \times d_a$ outputs, corresponding to the action dimension $d_a$ at each timestep. Input images are normalized with ImageNet statistics, and the model is trained with mean squared error on ground-truth actions. The inverse dynamics model is trained independently on the Bridge V2 dataset for 200k steps.

## C.1 VALIDATION VISUALIZATION OF LANGUAGE CONDITIONED VIDEO GENERATION MODEL

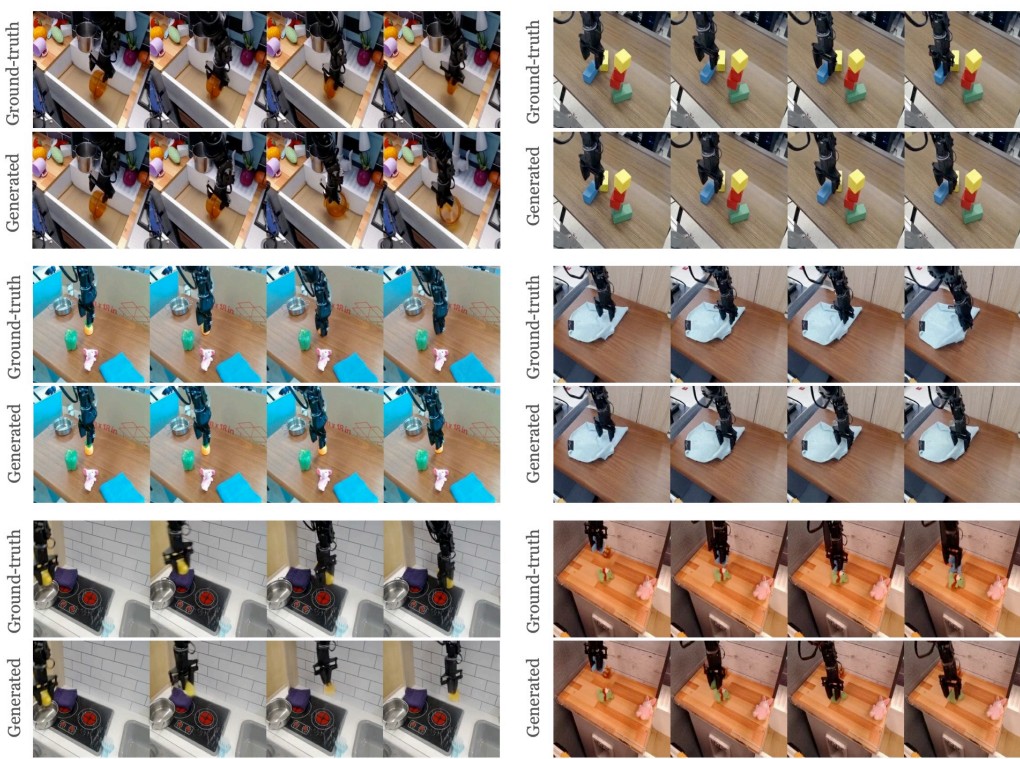

Figure 14: **Validation Visualization** of the language-conditioned video generation model on Bridge-V2. At inference, the model takes a UMT5-xxl instruction encoding and an initial frame, then predicts the next nine frames to complete the task.

# D ARCHITECTURE AND TRAINING DETAILS OF DIFFUSION POLICY

We followed the recipe of DexVLA (Wen et al., 2025) for training the diffusion policy. We load the Qwen2-VL-2B (Wang et al., 2024) backbone and the pre-trained control head, and perform an adaptation on BridgeV2 using LoRA (Hu et al., 2022), which inserts low-rank adapter matrices inside the backbone's attention and feed-forward blocks. These matrices along with the policy head are the only trainable modules in the adaptation stage. This preserves the backbone's general vision-language competence, and makes adaptation compute and memory efficient. We fine-tune the model with adapters on Bridge-V2 for 60k steps. During training, we rescale the actions to $(-1, 1)$ to match the diffusion target range.

DexVLA's policy head is trained as a denoising diffusion model with the standard $\epsilon$-prediction DDPM objective, i.e. at each update Gaussian noise is added to ground-truth action sequences at a randomly sampled diffusion step and the network is trained to predict that noise using an MSE loss. At inference, actions are generated with DDIM in a small number of steps, progressively denoising from a Gaussian initialization to a trajectory. We choose AdamW for the optimizer, using standard decoupled weight decay which applies decay to linear/attention weights, but exclude biases and LayerNorm parameters.

# E ADDITIONAL EXPERIMENTAL RESULTS

## E.1 ADDITIONAL RESULTS ON REAL-ROBOT VIDEOS

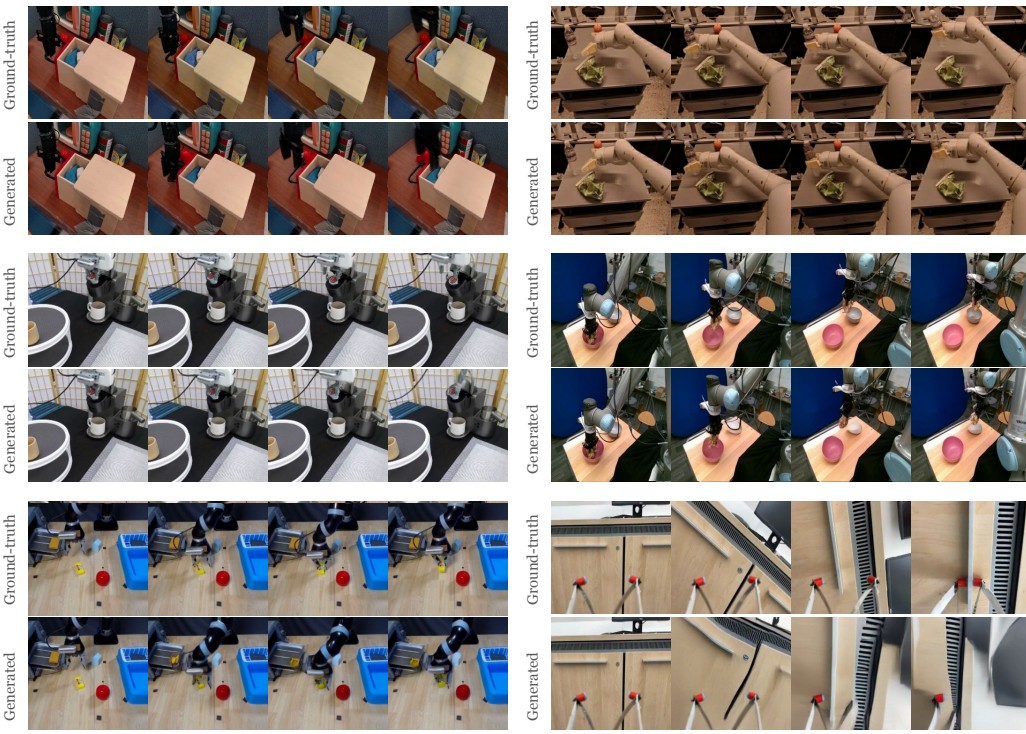

Figure 15: **Additional Qualitative Evaluation** of simulating actions from different robots. The world model generally generates the video that look very similar to the original video conditioned on the same actions that produced the original video in the real world.

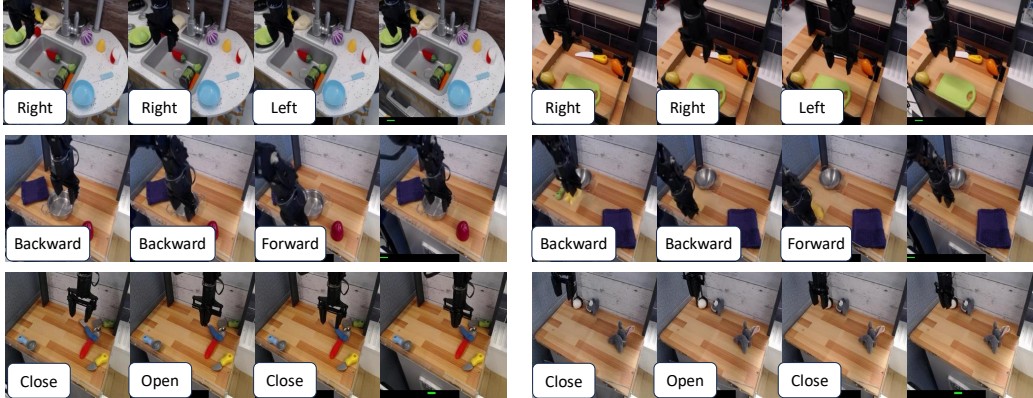

Figure 16: **Additional End-Effector Control Sweep on Bridge.** We simulate different gripper controls along different action dimensions corresponding to left-right, forward-backward, and gripper open-close. The world model generally generates videos that follow the actions.

### E.2 ADDITIONAL RESULTS ON GOOGLE ROBOT

To assess the generalizability of WorldGym, we performed rollouts with different policies on Google Robot. For our analysis, we chose a subset of tasks from the RT-1 dataset (Brohan et al., 2022). A partial score of 0.5 was assigned to a rollout if the robot attempted to reach the target location. OpenVLA again outperformed Octo and RT-1-X (see Table 4). However, in this environment Octo and RT-1-X are narrowly behind. The strong performance of RT-1-X might be due to a higher proportion of Google Robot trajectories than WidowX in its pretraining mix.

Table 4: **Policy rollouts on Google Robot** (RT-1 subset). OpenVLA outperforms RT-1-X and Octo, but by a smaller margin than on the Bridge dataset.

| Task | # Trials | RT-1-X # Successes | Octo # Successes | OpenVLA # Successes |
|------|----------|--------------------|------------------|---------------------|
| Close Bottom Drawer | 10 | 9 | 8.5 | 6 |
| Open Left Fridge Door | 10 | 4.5 | 3.5 | 4 |
| Pick Blue Chip Bag | 10 | 5 | 5.5 | 9 |
| Place Redbull Can Upright | 10 | 1.5 | 3 | 3.5 |

## E.3 DETAILED RESULTS ON THE OPENVLA BRIDGE EVALUATION TASKS

Table 5: **Detailed Bridge Evaluation Results** comparing RT-1-X (O'Neill et al., 2023), Octo (Octo Model Team et al., 2024), and OpenVLA (Kim et al.) on the Bridge evaluation suite of tasks from Kim et al.. Real-world task success rates are taken directly from (Kim et al.), WorldGym success rates are from rolling out policies within our world model.

| Task | # Trials | RT-1-X | | Octo | | OpenVLA | |
|------|----------|--------|--|------|--|---------|--|
| | | Real-world # Successes | WorldGym # Successes | Real-world # Successes | WorldGym # Successes | Real-world # Successes | WorldGym # Successes |
| Put Eggplant into Pot (Easy Version) | 10 | 1 | 1 | 5 | 1 | 10 | 7 |
| Put Eggplant into Pot | 10 | 0 | 0 | 1 | 2 | 10 | 6 |
| Put Cup from Counter into Sink | 10 | 1 | 3 | 1 | 3 | 7 | 9 |
| Put Eggplant into Pot (w/ Clutter) | 10 | 1 | 0.5 | 3.5 | 3.5 | 7.5 | 8 |
| Put Yellow Corn on Pink Plate | 10 | 1 | 3 | 4 | 6 | 9 | 9.5 |
| Lift Eggplant | 10 | 3 | 2 | 0.5 | 1.5 | 7.5 | 7.5 |
| Put Carrot on Plate (w/ Height Change) | 10 | 2 | 0.5 | 1 | 3 | 4.5 | 6 |
| Put Carrot on Plate | 10 | 1 | 0 | 0 | 1 | 8 | 4 |
| Flip Pot Upright | 10 | 2 | 3 | 6 | 1 | 8 | 5 |
| Lift AAA Battery | 10 | 0 | 1 | 0 | 0 | 7 | 4 |
| Move Skull into Drying Rack | 10 | 1 | 2 | 0 | 3 | 5 | 5 |
| Lift White Tape | 10 | 3 | 1 | 0 | 1 | 1 | 6 |
| Take Purple Grapes out of Pot | 10 | 6 | 5 | 0 | 2 | 4 | 4 |
| Stack Blue Cup on Pink Cup | 10 | 0.5 | 0 | 0 | 0 | 4.5 | 6 |
| Put {Eggplant, Red Bottle} into Pot | 10 | 2.5 | 0.5 | 4 | 5 | 7.5 | 9 |
| Lift {Cheese, Red Chili Pepper} | 10 | 1.5 | 2.5 | 2.5 | 2.5 | 10 | 10 |
| Put {Blue Cup, Pink Cup} on Plate | 10 | 5 | 1.5 | 5.5 | 5 | 9.5 | 8.5 |
| Mean Success Rate | | 18.5±4.0% | 15.5±3.4% | 20.0±5.3% | 23.82±4.3% | 70.6±6.1% | 67.4±4.9% |

We report the mean success rate across tasks with standard error (SE) computed as

$$\text{SE} = \frac{\text{sd}(r_1, \ldots, r_T)}{\sqrt{T}},$$

where $r_i$ is the per-task success rate and $T$ is the number of tasks.

## E.4 DETAILED RESULTS ON OOD IMAGE EVALUATION TASKS

Table 6: **Detailed Bridge OOD Image task results.** OpenVLA appears to be more robust across the different OOD settings of object generalization, distractions and classification.

| Category | Task | # Trials | RT-1-X # Successes | Octo # Successes | OpenVLA # Successes |
|---|---|---|---|---|---|
| Object Generalization | Pick up Orange (Carrot closer to Gripper) | 10 | 1 | 1 | 4 |
| Object Generalization | Pick up Orange (Orange closer to Gripper) | 10 | 3 | 4 | 9 |
| Object Generalization | Pick up Orange (Replace Carrot with Radish) | 10 | 1 | 4 | 10 |
| Distractor Robustness | Pick up Carrot (With Computer on side) | 10 | 6 | 7 | 9 |
| Distractor Robustness | Pick up Carrot (Computer closer to gripper) | 10 | 3 | 1 | 8 |
| Classification | Pick {Red, Blue} | 20 | 8 | 10 | 20 |
| Classification | Pick {Circle, Square} | 20 | 8 | 10 | 12 |
| Classification | Pick {Taylor Swift, Snoop Dogg} | 20 | 7 | 10 | 11 |

Table 7: **Policy rollout performance comparison in the presence of unrelated distractions.** OpenVLA is more robust to distractions over RT-1-X and Octo. However, all policies suffer significant performance drop in the presence of distractors.

| Task | # Trials | RT-1-X # Successes | Octo # Successes | OpenVLA # Successes |
|---|---|---|---|---|
| Put Eggplant into Pot (Easy Version) | 10 | 1 | 1 | 3 |
| Put Eggplant into Pot | 10 | 0 | 0 | 6 |
| Put Cup from Counter into Sink | 10 | 0 | 1 | 8 |
| Put Eggplant into Pot (w/ Clutter) | 10 | 0 | 0 | 4 |
| Put Yellow Corn on Pink Plate | 10 | 0 | 0 | 2 |
| Lift Eggplant | 10 | 0 | 1 | 7 |
| Put Carrot on Plate (w/ Height Change) | 10 | 0 | 0 | 2 |
| Put Carrot on Plate | 10 | 0 | 2 | 4 |
| Flip Pot Upright | 10 | 0 | 0 | 0 |
| Lift AAA Battery | 10 | 1 | 0 | 1 |
| Move Skull into Drying Rack | 10 | 3 | 0 | 5 |
| Lift White Tape | 10 | 0 | 0 | 4 |
| Take Purple Grapes out of Pot | 10 | 7 | 0 | 3 |
| Stack Blue Cup on Pink Cup | 10 | 0 | 0 | 4 |
| Put {Eggplant, Red Bottle} into Pot | 10 | 0 | 0 | 5 |
| Lift {Cheese, Red Chili Pepper} | 10 | 1 | 2 | 6 |
| Put {Blue Cup, Pink Cup} on Plate | 10 | 0 | 0 | 3 |
| Mean Success Rate | | 7.6±4.3% | 4.1±1.7% | 39.4±5.1% |

# F ABLATION STUDIES

## F.1 DATASET SIZE ANALYSIS

Table 8: **Dataset ablation.** Larger training dataset improves all three metrics comparing generated videos and ground-truth validation videos. ↑ means higher the better.

|  | Subset (Bridge V1) | Full (Bridge V2) |
|---|---|---|
| MSE ↓ | 0.015 | 0.010 |
| LPIPS ↓ | 0.131 | 0.073 |
| SSIM ↑ | 0.735 | 0.827 |

We measure MSE, LPIPS, and SSIM on generated videos from a model that is trained on less video data (Bridge V1 (Ebert et al., 2021)) and compare with a model that is trained on more data (Bridge V2 (Walke et al., 2023)). Table 8 shows that the model trained on more data leads to improvements in all three metrics.

## F.2 PARALLELISM EFFICIENCY ANALYSIS

Table 9: **Parallelism efficiency comparison.** Inference time for generating 40-frame video rollouts on an A100 GPU with different horizon lengths, demonstrating the efficiency gains from parallel frame denoising.

| Prediction Horizon | Time (s) |
|---|---|
| $h = 1$ | 93 |
| $h = 4$ | 33 |

Increasing the horizon from 1 to 4 frames achieves a 2.8× speedup. This is particularly useful for evaluating robot policies with differing action chunk sizes. For instance, OpenVLA (Kim et al.) predicts just a single action per frame, while Octo (Octo Model Team et al., 2024) predicts 4 actions per frame. Using the same world model checkpoint, we can improve the efficiency of rollout generations by matching the horizon to the policy's chunk size at inference time.

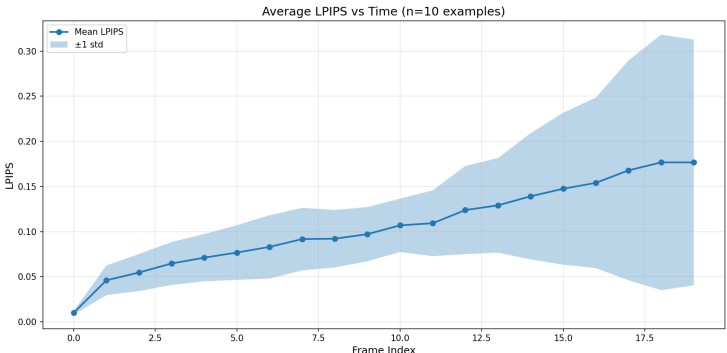

Figure 17: Mean LPIPS per rollout frame, averaged across 10 test-set rollouts. Lower is better. Visual error remains bounded throughout the rollout, staying below 0.2.

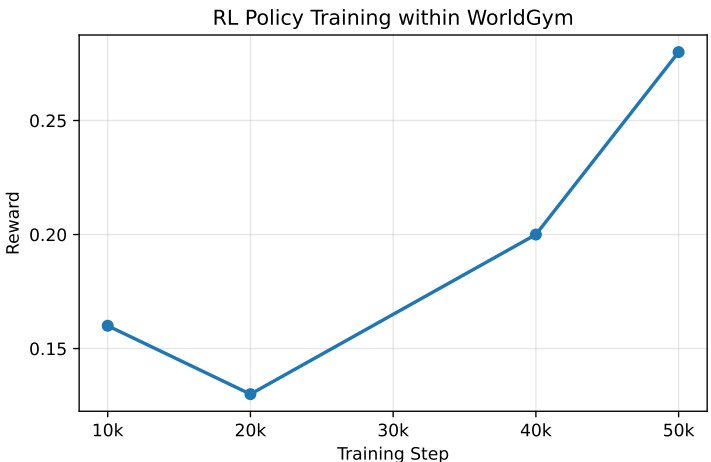

Figure 18: Mean success rate within WorldGym as a function of RL finetuning steps when finetuning OpenVLA with RL4VLA. Performance improves over the course of training.

## G ADDITIONAL EVALUATION METRICS

We additionally quantify long-horizon visual fidelity using LPIPS, computed between generated frames and ground-truth frames at each rollout timestep under a fixed action sequence. Figure 17 shows the mean LPIPS value (lower is better) at each frame index, averaged across 10 rollouts from the world model test set. The mean LPIPS remains below 0.2 throughout the full rollout, while increasing gradually over time as small prediction errors accumulate. This trend is expected, since minor discrepancies between the learned dynamics and the real world compound over longer horizons. Despite this drift, generated rollouts remain visually close to ground truth, and the world model still preserves strong correlation between ground-truth and simulated policy success rates. Future work can explore how further model scaling can reduce error accumulation over time.

## H REINFORCEMENT LEARNING WITHIN WORLDGYM

We also investigated whether world models can serve as effective training environments for reinforcement learning. As a preliminary study, we finetuned OpenVLA with RL inside WorldGym using the RL4VLA framework. In each episode, the policy is given a language instruction together with an initial frame from the Bridge V2 training set, then performs a rollout inside the world model while receiving reward from a VLM-based grader. We evaluate the resulting policy within WorldGym using initial frames and language instructions drawn from the Bridge V2 test set, and report mean success rate.

Figure 18 shows that policy performance improves with additional RL finetuning. Although these results are still preliminary, they suggest that WorldGym may be a useful environment for policy improvement and motivate further study of reinforcement learning inside learned world models.

