# OpenReview forum: "WorldGym: World Model as An Environment for Policy Evaluation"
_ICLR.cc/2026/Conference — ICLR 2026 Poster_

### Official Review · Reviewer_vCB6 · 2025-10-28

**Soundness:** 3
**Presentation:** 3
**Contribution:** 3
**Rating:** 8
**Confidence:** 3

**Summary:**

This paper proposes a framework to train a robot control sequence conditioned diffusion model to produce images. The model serves as a virtual world where robot policies can be validated in manipulation tasks. The language model can be used to provide rewards for the evaluation. In the experimental part it is shown that the model provides evaluations that are consistent with real-world evaluations. At its best the model can be used in training better models for real world tasks.

**Strengths:**

Since the diffusion models can provide realistic looking images and through language (or rather tokens) can be used to condition image generation, it is an interesting path of research for robot learning to use such models as components in robot learning. This idea is not new, but the implementation of this work is rather good and it can provide interesting avenues for the future research. I also believe it could be useful for offline RL.

**Weaknesses:**

**Major:**

**Moderate:**

 - The qualitative example sequences in Figure 2 are so short that it is barely visible something happens - a longer sequence from task start to task finish would be more interesting to see

 - More details about how much training data is needed to train the model. This could be in the terms of how many hours of video and how many episodes used, how the position of camera (view) affects the results (how much variation was allowed) => for the camera-ready you could add even more discussion about the limitations observed

 - Also some details about the computing setup and training times so that we know if this is doable in a research laboratory

 - Some example images of the out-of-dataset tasks would be interesting to see (i.e. how different they actually are)

**Minor:**

**Questions:**

I think your work is good and everything is explained in the level the results are replicable. I wish the code and pre-trained models are published since I am sure there are many technical details missing but important for those who want to replicate them.

I am sure the model has limitations, but I also find that it can be used in coarse evaluation of policies trained and perhaps to debug errors during development. I find it valuable and therefore important to be published.

It would be interesting to see how well the diffusion model can generalize given more training data (diversity + amount) and is there some kind of correlation with the policy i.e. could it be used to predict how much training data is needed to be able to learn a policy. And does diffusion model training need more data than learning a well working policy?

---

> ### Author Response · Authors · 2025-11-21
>
> Thank you for recognizing the value of this work! Here we hope to address the concerns you mentioned.
>
> ## Qualitative example sequences
>
> To view longer qualitative example sequences in GIF format, please open `video_html/rollout_videos.html` from the supplementary materials and scroll down to the Evaluation Rollouts, where you can see full policy rollouts in the world model.
>
> ## Training data
>
> We train on about 100 hours of robot data from the Open X-Embodiment dataset. We always use the primary over-the-shoulder observation from each dataset. Generalization can be greatly improved by increasing the dataset size and camera view diversity. This is an exciting area of future research.
>
> ## Training compute
>
> For the final policies used in this paper, we train on 4 H100s for 2-3 days, making this policy quite accessible to research.
>
> ## Out-of-dataset tasks
>
> For examples of OOD start frames, see Figure 8, 9, 10, and 12.
>
> ## Question: How does increased training data affect world model generalization?
>
> We expect the world model to generalize better with increased size and diversity of training data, and this is a direction which we are currently exploring!
>
> ## Question: Does world model training require more data than policy training?
>
> This is an exciting open research question. One interesting note is that world models need to be trained on both policy successes and failures to learn realistic dynamics, whereas behavior cloning policies are typically trained on successful trials only.
>
> ## Question: Will the code and pretrained models be open sourced?
>
> We share your excitement about using world models to improve policy development! The code and pretrained models will be open sourced and will be linked in the paper upon release (also see the code included in the supplementary materials).

---

### Official Review · Reviewer_tKDH · 2025-10-31

**Soundness:** 3
**Presentation:** 3
**Contribution:** 4
**Rating:** 6
**Confidence:** 3

**Summary:**

This paper introduces WorldGym, a framework that uses a video-based world model as an environment for evaluating robot policies. The model predicts future visual outcomes from actions and uses GPT-4o as a semantic reward evaluator, showing strong correlation with real-world policy performance.

**Strengths:**

The paper introduces a clear and innovative use of video diffusion world models for policy evaluation instead of training, which reframes how offline policy analysis can be done without physical robots. The experiments show consistent correlation between simulated and real results, maintain relative performance rankings across models, and include OOD tests that reveal model weaknesses. Technically, the framework is efficient, combining causal temporal attention and adaptive horizon prediction to support different policy granularities while keeping inference cost reasonable.

**Weaknesses:**

I believe a more convincing way to show WorldGym’s effectiveness would be to perform reinforcement learning with WorldGym as the environment and then test the resulting policy in simulation or the real world, but the paper doesn’t do that.

**Questions:**

Have the authors attempted to perform reinforcement learning using WorldGym as the training environment? If not, are there plans to test such sim-to-sim or sim-to-real transfer in future work?

---

> ### Author Response · Authors · 2025-11-21
>
> Thank you for your review and for recognizing the value of WorldGym for policy evaluation. Here we aim to address your question about using WorldGym as a training environment for reinforcement learning.
>
> ## Significance of policy evaluation
>
> We want to reiterate that the focus of this paper is on policy evaluation, which is a significant problem area within robotics. As the community shifts towards training foundation models for robotics, development depends on large-scale benchmarking on a wide variety of tasks. Researchers need to quickly evaluate many model checkpoints in order to understand (1) the real-world performance of a policy as it trains to determine convergence and (2) the effects of many model ablations on task performance, such as different model architectures and dataset mixtures. Furthermore, these benchmarks require many randomized trials for statistical significance and also a wide variety of tasks and initial environments in order to obtain a holistic measure of policies' robustness. Evaluation is so labor intensive that the top labs hire dozens of contractors just to monitor policy rollouts, while university students spend days carrying it out on their own.
>
> WorldGym provides an efficient solution: with just a single initial frame, researchers can test out a policy on a new task and environment in seconds. By curating the collection of initial frames to include randomized object locations and OOD visual settings, for example by using an image editing model as in our Section 4.3, researchers can obtain a policy success estimate over a wide variety of trials. Our framework makes this possible in under 1 GPU hour, without the manual labor of setting up and monitoring each trial, and with greater reproducibility as experiments can be repeated by just reusing the same start frame.
>
> ## Reinforcement learning within WorldGym
>
> We also wonder if world models can be used as training environments for reinforcement learning. We ran a side study this week to explore this possibility. We finetuned OpenVLA with RL inside of WorldGym, using the RL4VLA framework. In each episode, the policy is given a language instruction and an initial frame from the Bridge V2 training set, performing a rollout within the world model and receiving a reward from the VLM grader. We then evaluated the finetuned policy within WorldGym, using initial frames and language instructions from the Bridge V2 test set, reporting mean success rate. We found that the policy's performance on these tasks improves with further RL finetuning ([click here to see a plot of reward over time](https://i.imgur.com/7AxyjI0.png)).
>
> Given only one week, our results are still preliminary, but this is an exciting area of future research.
>
> [1] https://arxiv.org/abs/2505.19789

---

### Official Review · Reviewer_zRts · 2025-10-31

**Soundness:** 3
**Presentation:** 3
**Contribution:** 3
**Rating:** 6
**Confidence:** 4

**Summary:**

The paper proposes WorldGym: a learned, action-conditioned video world model used as a unified environment to offline-evaluate robot policies. From a single real initial frame and a policy’s action chunk, the model predicts future frames; a VLM grader turns those into success scores. A key systems choice aligns the prediction horizon with each policy’s action-chunk size to reduce wasted rollout compute. Experiments report strong sim-to-real correlation of success rates, preserved policy rankings across families/sizes/checkpoints, and targeted OOD probes that reveal failure modes. The setup is framed as OPE with learned dynamics and learned reward from videos.

**Strengths:**

1. Originality: reframes policy evaluation as “rollout in one learned world” rather than per-task simulators; leverages the one-world prior and diverse training data.
2. Practicality: one real frame + actions, no hand-coded simulators; horizon–chunk alignment is a clean trick that supports mixed policies while saving compute.
3. Clarity: the OPE formulation and rollout protocol are easy to follow; model/policy interfaces are explicit.
4. Significance: high sim-to-real correlation and preserved rankings make this a plausible tool for fast model selection; OOD edits provide actionable diagnostics.

**Weaknesses:**

1. VLM reward calibration is under-analyzed: the proposed VLM grader is central, but the paper does not show reliability audits (human agreement, prompt/temperature sensitivity, temporal credit). The authors should add thorough calibration and robustness studies.
2. Dynamics fidelity over long horizons is not quantified: the work shows plausibility, but does not report compounding-error metrics (e.g., FVD/LPIPS vs time, controllability under action perturbations). The authors should measure error growth and contact realism.
3. Statistics are light: correlations are promising, but the paper does not provide per-task CIs, bootstrap uncertainty, or stress tests under harder domain shifts (lighting, camera pose, clutter). Add stronger stats and broader shifts.
4. Single-frame initialization sensitivity is unclear: the paper does not analyze bias/leakage from the initial frame. Evaluate with occlusions/crops/background swaps to quantify sensitivity.
5. Efficiency claims are not profiled: horizon–chunk alignment is appealing, but the paper does not show throughput/latency/memory vs baselines across GPUs and chunk lengths. Provide wall-clock profiles and ablations (e.g., with/without diffusion forcing).
6. Related-work positioning lacks shared benchmarks: there is no head-to-head vs closest OPE/world-eval baselines on common tasks. Add small shared batteries and ranking-agreement metrics.
7. OOD analysis is narrow: findings hinge on a few edit types; generality across textures, shapes, adversarial overlays is unknown. Expand OOD suite and report policy-wise degradations with uncertainty.

**Questions:**

Same in weakness.

---

> ### Author Response · Authors · 2025-11-21
>
> Thank you for your detailed review. We would like to address each of the concerns you have below.
>
> ## 1: VLM reward calibration is under-analyzed.
>
> Justifying VLM reward reliability is indeed an important step. Please see Appendix B.2, Table 3, where we validate the VLM success predictions via a confusion matrix. We find a high true positive rate of 0.81 and a low false positive rate of 0.03. We saw in our initial experiments that the VLM-as-reward was not always reliable. So, we introduced majority voting, where we caption the rollout 5 times with the same prompt and take the score which receives the most votes. We found this to significantly improve the VLM's reward prediction accuracy.
>
> ## 2: Dynamics fidelity over long horizons is not quantified.
>
> Quantifying LPIPS over time for a fixed sequence of actions is a great suggestion which we considered as well. [Click here to see a plot of LPIPS](https://i.imgur.com/ZPzksbP.png) (lower is better) over time, averaged across 10 rollouts from the world model's test set. We can see that the mean LPIPS value remains below 0.2 throughout the entire rollout, though steadily increases after the initial frame due to natural differences between the world model and the real world. Despite this, rollouts remain mostly visually similar to ground truth (see top of `video_html/rollout_videos.html` in supplementary material), and the world model offers high correlation between ground truth and simulated policy success rates.
>
> ## 3: Statistics.
>
> Please note that we include standard error bars for all success rates. See Figure 4b and Table 5.
>
> ## 4: Initial frame sensitivity.
>
> We would love to address this question but we are unsure what the reviewer means by sensitivity, bias, and leakage here. The world model predicts future frames conditioned on the start image, and so naturally the future frames predictions are highly dependent on the initial frame and must preserve its general scene composition in order to achieve a low prediction error. Could you provide additional clarification on this question?
>
> ## 5: Efficiency profiling.
>
> Please see Table 9 in Appendix F.2 where we profile the inference efficiency for different chunk lengths on 1xA100 GPU.
>
> ## 6: Related work lacks shared benchmarks.
>
> Shared benchmarks would be a great way to evaluate the progress of world models for policy evaluation. At the time of submission however, the only existing world model policy evaluation method was WorldEval [1], which only trains the world model on a custom evaluation environment which differs significantly from the Bridge evaluation tasks proposed by OpenVLA. The next closest work is SIMPLER [2], a software simulator-based offline policy evaluation method which requires manual programming and careful green-screening to set up a new task. The only task which SIMPLER has implemented which is directly comparable to our eval tasks in "Close Drawer" task on the Google Robot (see Table IV in SIMPLER and Table 4 in WorldGym). The relative success rates are the same across SIMPLER and WorldGym, with both finding that RT-1-X outperforms Octo. SIMPLER additionally implements the "Put Carrot on Plate" task, however it is on a wooden table environment, while we use the initial frames from OpenVLA's evaluation, which uses a kitchen sink environment. Computing the Pearson correlation between real-world and simulated task performance, we find that SIMPLER achieves r=0.57 (see Table V in SIMPLER) while WorldGym achieves r=0.94 (see Table 5 in WorldGym) for this task, with the caveat that SIMPLER evaluates RT-1-X, Octo-Base, and Octo-Small, while we evaluate RT-1-X, Octo-Base, and OpenVLA.
>
> ## 7: OOD analysis is narrow.
>
> Performing a wide array of OOD experiments is a great suggestion to further benchmark policy robustness. Please note that we already report policy-wise degradations with error bars for the adversarial distractor OOD experiment (see Figure 13), where we find that OpenVLA degrades the least. This OOD task presents the policies with a diverse challenge because objects with random shapes, sizes, and colors are added to the scene (see Figure 12 for some examples). We were also curious about many different OOD edit types, and so we additionally explored modifying object shape and color (Figure 9), colored paper classification (Figure 8), the effects of adversarial distractor computers (Figure 10, left), and advanced classification tasks like shape classification, celebrity facial recognition, and cat vs dog classification (Figure 10, right). Furthermore, we explored additional OOD edits not included in the paper, such as altering the camera viewpoint for the Bridge sink environment ([click here to view a GIF of OOD viewpoint rollout](https://i.imgur.com/kHCNlNh.gif)).
>
> [1] https://arxiv.org/abs/2505.19017
>
> [2] https://arxiv.org/abs/2405.05941

---

### Official Review · Reviewer_UNV5 · 2025-11-01

**Soundness:** 4
**Presentation:** 3
**Contribution:** 3
**Rating:** 6
**Confidence:** 4

**Summary:**

This paper introduces WorldGym, a world-model-based environment for evaluating robot control policies without physical deployment. The approach uses an autoregressive, action-conditioned video generation model to simulate robot interactions, with a VLM (GPT-4o) providing task success evaluation. The authors demonstrate strong correlation (r=0.78) between world model success rates and real-world performance across three VLA policies (RT-1-X, Octo, OpenVLA) on Bridge manipulation tasks. WorldGym enables efficient evaluation on OOD tasks and environments through language/image modifications, revealing interesting policy failure modes.

**Strengths:**

- WorldGym proposes a way to address a genuine need for safe, reproducible, and cost-effective policy testing before real-world deployment.
- WorldGym shows impressive correlation between simulated and real-world success rates. It also does strong empirical validation.
- The single world model generalizes across diverse tasks and environments.
- The paper is well-written, the problem motivation is compelling, and the approach is clearly described.

**Weaknesses:**

- The paper does provide quantitative metrics on world model prediction quality.
- The world model still shows physics-inplausible predictions, as shown in Figure 14. The paper also does not propose any method to help the video model learn real-world physics.
- The paper does not compare with baselines like building digital twins to do policy evaluation.
- The paper does not show the computational requirements and the inference speed of the world model.
- The paper does not show qualitative results on the simulation and real-world correlation. For example, can authors provide the visualizations of model success and failure cases to see if WorldGym has similar visual renderings compared with the real world? Or providing a fixed set of action replay in both the real world and WorldGym is also meaningful to see the world model gap.

**Questions:**

See weaknesses.

---

> ### Author Response · Authors · 2025-11-21
>
> Thank you for your detailed review. Here we address each of the concerns you have for this work.
>
> ## 1: The paper does provide quantitative metrics on world model prediction quality.
>
> Quantitative evaluations of the world model are crucial to understanding its performance. Please see Appendix F.1, Table 8, where we provide quantitative metrics, namely MSE, LPIPS, and SSIM, on the world model's prediction quality. Given an initial frame and a sequence of actions from the validation set, we roll out those actions in the world model, and compare the generated video to the ground truth rollout video. Table 8 shows that increasing the training dataset size results in an improvement across all three of these quantitative metrics.
>
> ## 2: Physics-implausibility of the world model.
>
> It is true that generating highly realistic object interactions remains challenging for the world model, as we note in the introduction and conclusion, and improving the physical realism of the model is an exciting area of future research. We want to highlight that the focus of this paper is on policy evaluation. Despite some inaccurate object interaction simulation, WorldGym still achieves high aggregate correlation with real-world policy performance (r=0.78), making it a useful tool for distinguishing between policies without the manual labor of monitoring real world robot rollouts. To improve physical realism of world models, future work can scale the training dataset size. The Open X-Embodiment dataset we train on consists of just a few hundred hours of robot interaction. By collecting larger robotics datasets, or by learning from large scale internet videos which contain object interactions, future world models' prediction quality can be greatly improved.
>
> ## 3: Comparison with digital twins as a baseline.
>
> Comparing with software-based digital twins is a great suggestion which we considered as well. Existing digital twin works which were released prior to this paper's submission include Real-is-Sim [1] and SIMPLER [2]. Real-is-Sim creates a gaussian-splat based digital twin but requires rolling out each simulated action on a real robot to update the digital twin's state at each timestep, while we aim for offline policy evaluation without a physical robot. SIMPLER allows for offline policy evaluation in a software simulator, but requires manual programming and careful green-screening to set up a new task. The only task which SIMPLER has implemented which is directly comparable to our eval tasks in "Close Drawer" task on the Google Robot (see Table IV in SIMPLER and Table 4 in WorldGym). The relative success rates are the same across SIMPLER and WorldGym, with both finding that RT-1-X outperforms Octo. SIMPLER additionally implements the "Put Carrot on Plate" task, however it is on a wooden table environment, while we use the initial frames from OpenVLA's evaluation, which uses a kitchen sink environment. Computing the Pearson correlation between real-world and simulated task performance, we find that SIMPLER achieves r=0.57 (see Table V in SIMPLER) while WorldGym achieves r=0.94 (see Table 5 in WorldGym) for this task, with the caveat that SIMPLER evaluates RT-1-X, Octo-Base, and Octo-Small, while we evaluate RT-1-X, Octo-Base, and OpenVLA.
>
> ## 4: Computational requirements and the inference speed of the world model.
>
> We discuss the inference speed of the model in Appendix F.2 (referenced in Section 3.1.2). We find that a 40-frame rollout takes 93 seconds with horizon=1 and 33 seconds with horizon=4 on a single A100.
>
> ## 5: Qualitative results on the simulation and real-world correlation.
>
> Qualitative results of the task rollouts can be found in Figure 5 and in GIF format in supplemental materials (open `video_html/rollout_videos.html`, see section "Evaluation Rollouts For Bridge Data"). We also have real-world rollout recordings for OpenVLA on these tasks – see [the attached GIF linked here](https://i.imgur.com/u7ZjGOH.gif) for a side-by-side comparison. Note that the actions taken by the policy may be slightly different due to differences in visual observations in the real world vs world model.
>
> Comparing a fixed sequence of actions in the world model and the real world is a great way to measure the prediction accuracy of the world model – please see the "Rollout Examples" in supplementary materials `video_html/rollout_videos.html` for side-by-side comparisons across robots.
>
> [1] https://arxiv.org/abs/2504.03597
>
> [2] https://arxiv.org/abs/2405.05941

---

### Meta-Review · Area_Chair_FxYb · 2026-01-06

**Summary:**

This paper proposes WorldGym, a world-model-based environment for evaluating robot policies. It embraces an autoregressive format that can be used to assess policies without actual deployment. Experiments show the evaluation results are highly correlated with real-world evaluation performance. WorldGym can help with OOD tasks evaluation and reveal rare failure modes.

**Reviewer Concerns:**

Partially resolved concerns:
1. Comparison with physically grounded simulations.
2. Some of the core components needs verification such as VLM grader.
3. Training details.

Some concerns might still need further attention:
RL on WorldGym. While a side study is provided, an extensive and rigorous study can further demonstrate the capability of this WorldGym.

**Reviewer Scores:**

6 6 6 8

---

### Decision · Program_Chairs · 2026-01-26

Accept (Poster)